# Modulated Variational auto-Encoders for many-to-many musical timbre transfer

## Abstract

Generative models have been successfully applied to image style transfer and domain translation. However, there is still a wide gap in the quality of results when learning such tasks on musical audio. Furthermore, most translation models only enable *one-to-one* or *one-to-many* transfer by relying on separate encoders or decoders and complex, computationally-heavy models. In this paper, we introduce the Modulated Variational auto-Encoders (MoVE) to perform *musical timbre transfer*. We define timbre transfer as applying parts of the auditory properties of a musical instrument onto another. First, we show that we can achieve this task by conditioning existing domain translation techniques with Feature-wise Linear Modulation (FiLM). Then, by replacing the usual adversarial translation criterion by a Maximum Mean Discrepancy (MMD) objective, we alleviate the need for an adversarial objective. This allows a faster and more stable training along with a controllable latent space encoder. By further conditioning our system on several different instruments, we can generalize to *many-to-many* transfer within a single variational architecture able to perform multi-domain transfers. Our models map inputs to 3-dimensional representations, successfully translating timbre from one instrument to another and supporting sound synthesis from a reduced set of control parameters. We evaluate our method in reconstruction and generation tasks while analyzing the auditory descriptor distributions across transferred domains. We show that this architecture allows for generative controls in multi-domain transfer, yet remaining light, fast to train and effective on small datasets[1].

## 1 Introduction

Music information can be analyzed in many forms, each of which conveys different specificities over musical qualities. Among these, *timbre* is the set of properties that distinguishes one instrument from another playing at the same pitch and loudness. Timbre has become a core concept in music composition since the 19[th] century (McAdams (2013)). It has been studied using human dissimilarity ratings to construct *timbre spaces*, which exhibit the perceptual relationships between instruments (Grey (1977)). However, these spaces are not invertible to the signal domain and do not generalize to new examples (McAdams et al. (2006)). The heavy reliance on hand-crafted audio descriptors to analyze timbre perception altogether leads to a lack of established models to understand and generate timbres (McAdams (2013)). Moreover, the specific nature of music tasks requires tailored evaluation principles that are yet to be ascertained (Jaffe (1995)).

Recent advances in *generative models* open alternative avenues to analyze highly dimensional data and tackle complex subsequent tasks. Amongst these, the idea of *style transfer* (Gatys et al. (2015)) has recently gained a flourishing interest. This approach allows to modify the stylistic features of an image while preserving its overall content and led to the more generic question of *domain translation*. In the recent UNsupervised Image-to-image Translation (UNIT) model, Liu et al. (2017) proposed to learn a shared latent space with a Variational Auto-Encoder (VAE) and translate between different data domains with an adversarial criterion. However, specific properties of the generation cannot be controlled and that discriminative objective might lead to an unstable and longer training. Here, we first extend this approach to musical transfer while improving it by introducing Modulated Variational auto-Encoders (MoVE) that offer control over the generation through conditioning.

---

[1] Audio examples, source code and animations available at https://github.com/anonymous124/iclr2019MoVE

Furthermore, by replacing the discriminative networks by a Maximum Mean Discrepancy (MMD) objective, we alleviate the need for an additional adversarial training specific to each domain.

Although UNIT provides a powerful framework, it only applies to *one-to-one* transfer. This implies that a different model has to be trained for each pair of domains. To mitigate this issue, Choi et al. (2017) proposed StarGAN which performs *many-to-many* transfer between several domains. However, it relies solely on Generative Adversarial Networks (GANs) and do not learns an implicit task representation to interact with. In the music realm, Mor et al. (2018) proposed Universal Music Translation (UMT), which does not use GANs. Although it enables translation across multiple complex audio domains, this method requires to learn a separate decoder for each domain, which leads to a prohibitive training time. In contrast to these methods, we show that MoVE can be further conditioned on domain information and generalizes to *many-to-many* transfer with a single encoder and decoder architecture able to perform multi-domain transfer. The resulting models are rather lightweight and fast to train while effective on a moderate amount of examples.

Here, we define *timbre transfer* as applying a variable part of the auditory properties of a musical instrument onto another. We circumvent the lack of definition for timbre by considering each instrument as a separate domain that maps onto a common latent representation. We further address the crucial need for interactivity and control in creative applications such as audio synthesis. Accordingly, our method yields 3-dimensional latent spaces that can be explored and controlled through high-level explicit variables such as pitch and octave values. This supports sound generation with smoothly evolving timbre qualities and complex domain transfers from a reduced set of parameters. Finally, we analyze traditional audio descriptor distributions when transferring between multiple domains or decoding across latent dimensions to demonstrate the generative capacities of our model.

## 2 Related works

**Style transfer and image translation.** In computer vision, style transfer (Gatys et al. (2015)) has been proposed to generate images that preserve the content from a source image but feature stylistic qualities belonging to another target image. Although this technique provides compelling results, it operates on local textural information and fails to capture higher-level semantic properties of the style. Further research has been carried to address this question of *domain translation*, first proposed by Isola et al. (2017). In the fully supervised setting, this translation would require paired samples. However, such datasets are scarce and the concept of existing samples exactly matching the translation task is restrictive from a generative perspective. In the UNIT approach (Liu et al. (2017)), the underlying assumption is that two hypothetically matching samples should map onto the same point in a shared latent space. Hence, translation is achieved by partially weight-shared VAEs in order to map the two separate domains to a common latent representation. Learning is performed with an auxiliary pair of adversarial discriminators which push translated samples to match the distributions of their respective domains. An additional *cycle-consistency* objective (Zhu et al. (2017)) reinforces the shared learning by ensuring that translated samples can be retrieved back to their original domains. However, this architecture can only operate on single domain pairs.

In order to provide *many-to-many* translations, Choi et al. (2017) proposed to replace weight-sharing by conditioning a single GAN. This allows to train on multiple domains simultaneously, while enabling control over the generative process. However, the authors evaluate only on highly similar domains (eg. face attributes). Furthermore, this approach relies solely on GANs, which are notoriously difficult to train, prone to lack full support over the data (Grover et al. (2017)) and do not provide a latent space encoder.

Recently, Feature-wise Linear Modulation (FiLM) has been proposed to improve conditioning by learning conditional bias and scaling throughout a network (Perez et al. (2017)). This method was successfully used in image stylization (Ghiasi et al. (2017)) where adaptive modulation conditioned on a style image is applied after each intermediate instance normalization. Here, we show that by relying on FiLM layers for domain conditioning, we can perform *many-to-many* domain translation with a single VAE architecture, as depicted in Fig. 1(c). Moreover, by using a MMD criterion, we alleviate the need for GANs or specific adversarial discriminators. Hence, we obtain an unsupervised, lightweight and easy to train model with a general and controllable latent space.

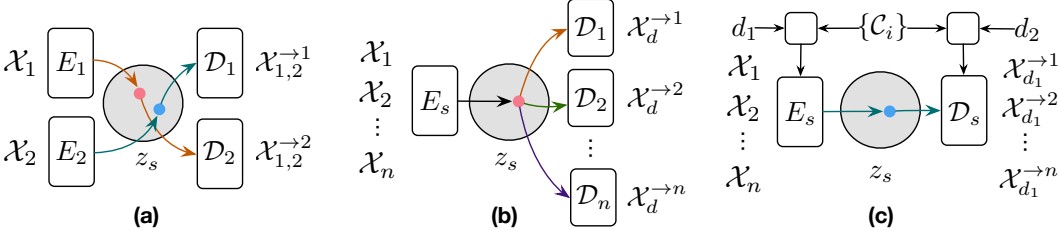

Figure 1: Different approaches to domain translation. (a) *One-to-one* transfer models such as UNIT are restricted to domain pairs. (b) *One-to-many* transfer models such as UMT require to train a different decoder for each domain. (c) Our proposed many-to-many transfer model (MoVE) allows to perform multi-domain transfer with a single encoder and decoder, while providing control over the generation with external high-level conditioning variables.

**Audio translations.** Recent applications of generative models to audio have shown promising results, notably supported by solutions to efficiently generate waveforms such as Wavenet (Van Den Oord et al. (2016)) or SampleRNN (Mehri et al. (2017)). Most of these proposals target voice signals and there is still a large gap when addressing musical data. Some approaches have tackled musical style transfer (Verma & Smith (2018)). However, as pointed by Dai et al. (2018), musical style is a multimodal and multi-scale notion, which implies a variety of underlying factors. Specifically for domain translation, Mor et al. (2018) proposed a Universal Music Translation (UMT) network that globally translates musical recordings between different genres and instrument domains. Using a single Wavenet encoder and separate decoders for each domain, this approach is able to transform a given melody so that it is played by different instruments. By design, this method requires to train a specific Wavenet decoder for each of the target domain. It does not provide control over audio synthesis and the learned representation does not allow direct visualization nor transfer of only specific parts of timbre attributes. Hence, it does not enable informed generative processes, musical interaction and creativity. Our proposal targets 3-dimensional latent spaces supporting timbre transfer and continuous synthesis paths with explicit control over musical attributes.

## 3 MUSICAL TIMBRE TRANSFER

*Musical timbre* can be defined as the *set of auditory qualities* that distinguishes two instruments playing the same note at the same loudness. Seminal studies relying on human dissimilarity ratings provided an interesting step towards understanding music perception (McAdams et al. (1995)). However, the ordination techniques used yield non-invertible and fixed timbre spaces. Hence, they do not support audio synthesis nor do they provide a way to manipulate timbre structures. Signal processing techniques have also been developed to process and alter timbre. However, these rely on complex analysis schemes that decompose sounds into large sets of parameters (Serra (1997)), precluding intuitive control over the audio synthesis process.

Here, we propose to use generative models in order to perform musical timbre transfer. In order to circumvent the complexity of defining timbre, our underlying hypothesis is that each instrument defines a timbral *domain,* which contains all style qualities that shape its identity. Musical timbre transfer can be achieved by transforming a certain amount of the auditory features of a musical instrument according to another (eg. like playing a saxophone with a bow). Transferring all timbre properties of an instrument leads to *domain translation*, while partial modification of these amounts to *style transfer*. Furthermore, our goal is to obtain a controllable model that can be used for creative purposes. Hence, we aim to obtain 3-dimensional latent spaces along with high-level musical parameters that enable human interaction and control over the generation.

This type of transfer can be performed in several ways, as depicted in Fig. 1. First, *one-to-one* transfer models such as UNIT map samples from a given pair of domains to a shared latent space. By learning separate layers and weight-shared layers in both the decoder and encoder, domain translation can be assessed through adversarial discriminators. We first adapt this model to timbre transfer and show that we can alleviate the need for GAN training by using an alternative MMD objective.

It leads to a faster and more stable learning that we further enhance by modulating shared layers with FiLM layers (Perez et al. (2017)) on pitch and octave. This provides an explicit control over generation, altering or not the pitch regardless of timbre. The *one-to-many* transfer models (UMT) allow to work with multiple domains but require to learn a different decoder for each. This leads to a more complex and longer training and reduces the generalization ability of the model gained through multi-task learning. Here, we show that our proposed Modulated Variational auto-Encoder (MoVE) allows to perform *many-to-many* transfers with a single VAE simultaneously processing all domains. The success of our solution relies on an efficient domain conditioning, together with external control variables, performed through FiLM layers acting on the whole network. This solution offers a greater generalization power by jointly learning all transfer tasks within a single architecture. The resulting latent space successfully models joint and conditional distributions over several instrument domains. This also enables control with semantic labels, while providing interactive 3-dimensional spaces to synthesize novel tones from a reduced set of control parameters.

## 3.1 ONE-TO-ONE TRANSFER

Our *one-to-one* transfer model is based on an architecture similar to UNIT (Liu et al. (2017)) where the core idea is to learn a latent space that is shared between two domains $\mathcal{X}_1$ and $\mathcal{X}_2$. Based on samples $x_1 \in \mathcal{X}_1$ and $x_2 \in \mathcal{X}_2$, we aim to model the joint distribution $p_{\mathcal{X}_1, \mathcal{X}_2}(\mathrm{x}_1, \mathrm{x}_2)$ over the two domains. By learning domain-specific encoders $E_1$ and $E_2$, matching samples drawn from each marginal distribution $p_{\mathcal{X}_1}(\mathrm{x}_1)$ and $p_{\mathcal{X}_2}(\mathrm{x}_2)$ should map onto the same latent code $\mathrm{z} = E_1(\mathrm{x}_1) = E_2(\mathrm{x}_2)$. Equally, any latent code can be decoded back to any of the two domains $d \in \{1, 2\}$ by learning appropriate decoders $\mathrm{x}_d^* = D_d(\mathrm{z})$. A paired VAE implements this assumption through separate domain-specific layers $\{e_d ; d_d\}$ alternated with weight-shared ones $\{e_{\mathrm{ws}} ; d_{\mathrm{ws}}\}$. The full encoders and decoders are defined by the composition of both parts $E_d = e_{\mathrm{ws}} \circ e_d$ and $D_d = d_d \circ d_{\mathrm{ws}}$.

Each VAE is trained with a reconstruction loss on its own domain, by approximating the intractable latent conditional $p(\mathrm{z}|\mathrm{x})$ with a parametric encoding network $q_\phi(\mathrm{z}|\mathrm{x})$ with $\phi \in \Phi$. In comparison to UNIT, we both use a Gaussian encoder $q_\phi$ and decoder $p_\theta$ so that $\mathrm{z} \sim q_\phi(\mathrm{z}|\mathrm{x}) = \mathcal{N}(\mu_\phi(\mathrm{x}), \sigma_\phi(\mathrm{x}))$ and $\mathrm{x} \sim p_\theta(\mathrm{x}|\mathrm{z}) = \mathcal{N}(\mu_\theta(\mathrm{z}), \sigma_\theta(\mathrm{z}))$. Training the model amounts to optimize $\{\theta ; \phi\}$ on the Evidence Lower Bound Objective (ELBO), defined as a Negative Log-Likelihood (NLL) term on the output prediction error and a $\beta$-weighted Kullback-Leibler Divergence (KLD) term that assesses the error from the approximate latent density against the intractable true posterior distribution.

$$\mathcal{L}_{\theta,\phi}^{rec.} = \mathbb{E}_{q_\phi(\mathrm{z})}[\log p_\theta(\mathrm{x}|\mathrm{z})] - \beta * D_{\mathrm{KL}}[q_\phi(\mathrm{z}|\mathrm{x}) \| p_\theta(\mathrm{z})] \tag{1}$$

This inference objective allows to learn structured low-dimensional and invertible representations of the data, while disentangling generative factors in the encoded variables (Higgins et al. (2016)).

Translation is performed by switching domains between the encoding and decoding stages eg. $\mathrm{x}_{1 \to 2} = D_2 \circ E_1(\mathrm{x}_1)$. However, there is usually no matching sample $\mathrm{x}_2^*$ that could allow to perform the optimization of the reconstruction error $\mathcal{L}_{\theta_2,\phi_1}^{1 \to 2} = err(\mathrm{x}_{1 \to 2} \| \mathrm{x}_2^*)$. To circumvent this challenge, UNIT relies on GANs to discriminate the generated translations against the target data distributions they model. However, this GAN criterion leads to a more complex and possibly unstable training process. Here, we show that we can efficiently replace the adversarial criterion by a differentiable distance measure on the probability distribution spaces. We minimize the Maximum Mean Discrepancy (MMD), a non-parametric kernel method (Gretton et al. (2012)), between the set of transferred samples $\mathrm{x}_{1 \to 2} \sim p_{\theta_2}(\mathrm{x}|\mathrm{z}, \phi_1)$ and a randomly sampled set from the target domain $\bar{\mathrm{x}}_2 \sim p_{\mathcal{X}_2}$

$$\mathcal{L}_{\theta_2,\phi_1}^{1 \to 2} = \mathrm{MMD}[\mathrm{x}_{1 \to 2} \| \bar{\mathrm{x}}_2] = \mathbb{E}_{\mathrm{x}, \mathrm{x}'}[k(\mathrm{x}, \mathrm{x}')] - 2 * \mathbb{E}_{\mathrm{x}, \bar{\mathrm{x}}}[k(\mathrm{x}, \bar{\mathrm{x}})] + \mathbb{E}_{\bar{\mathrm{x}}, \bar{\mathrm{x}}'}[k(\bar{\mathrm{x}}, \bar{\mathrm{x}}')]$$
$$\forall \{\mathrm{x}, \mathrm{x}'\} \in \mathrm{x}_{1 \to 2} \text{ and } \forall \{\bar{\mathrm{x}}, \bar{\mathrm{x}}'\} \in \bar{\mathrm{x}}_2 \tag{2}$$

where $k$ is a Radial Basis Functions (RBF) kernel $k(\mathrm{x}, \mathrm{x}') = \sum_{i=1}^{n} \exp^{-\alpha_i \|\mathrm{x} - \mathrm{x}'\|^2}$.

Reconstruction and translation objectives are jointly optimized with an extra circle-consistency (CC) criterion. It consists in encoding a translated sample back to the latent space and decoding it to its source domain so that $\mathrm{x}_{cc1} = D_1 \circ E_2(\mathrm{x}_{1 \to 2})$. Hence, this double translation should retrieve the initial sample and the reconstruction error can be optimized with a NLL loss.

$$\mathcal{L}_{\theta_1,\phi_2}^{cc1} = \mathbb{E}_{q_{\phi_2}(\mathrm{z}|\mathrm{x}_{1 \to 2})}[\log p_{\theta_1}(\mathrm{x}|\mathrm{z})] \tag{3}$$

Finally, the complete optimization objective is defined as

$$\mathcal{L}_{\theta,\phi}^{train} = \mathcal{L}_{\theta_1,\phi_1}^{rec1} + \mathcal{L}_{\theta_2,\phi_2}^{rec2} + \lambda_{\text{MMD}}(\mathcal{L}_{\theta_2,\phi_1}^{1\to2} + \mathcal{L}_{\theta_1,\phi_2}^{2\to1}) + \lambda_{\text{CC}}(\mathcal{L}_{\theta_1,\phi_2}^{cc1} + \mathcal{L}_{\theta_2,\phi_1}^{cc2}) \tag{4}$$

where $\lambda_{\text{MMD}}$ and $\lambda_{\text{CC}}$ allow to weigh the relative influence of different objectives. For the purpose of controllable musical timbre transfer, we further apply conditioning at the input of the weight-shared networks by concatenating one-hot encoded pitch classes and octaves. This pushes the shared encoder to structure note-agnostic features, while providing control over the generation.

## 3.2 MANY-TO-MANY TRANSFER

In order to alleviate the *one-to-one* limitation that requires a different training for each domain pair, we propose the single MoVE architecture as depicted in Figure 2. All layers are shared over the multiple domains processed, by learning a single modulated encoder $E_s$ and decoder $D_s$. Transfer is performed by switching the categorical condition between different instruments. Hence, the practical success of this method highly depends on the conditioning strategy, which must also retain the pitch and octave control. To do so, we use an input embedding that jointly maps these categorical conditions to dense vectors fed into FiLM generators. We replace each intermediate batch normalization with instance normalization and activation is followed by a FiLM modulation layer (conditional instance normalization). Biasing and scaling are either applied feature-wise for 1-dimensional activations or channel-wise after 2-dimensional feature maps. A different generator output is used for modulating each instance normalization layer depending on its shape. The MoVE model trains in reconstruction with the ELBO and in transfer with the MMD, which is separately computed for each instrument against each of the others.

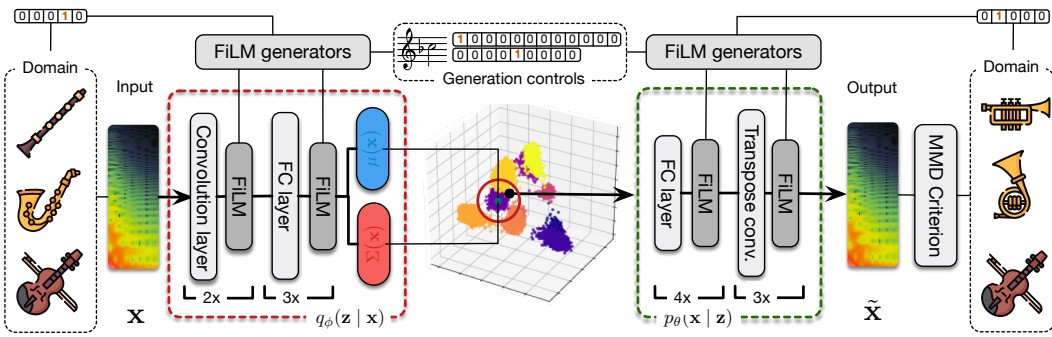

Figure 2: The Modulated Variational auto-Encoder (MoVE) provides a single architecture able to perform many-to-many transfer while controlling the generation with external parameters. Both the domain and control information are processed to modulate different layers of the architecture.

## 4 EXPERIMENTS

**Dataset.** In order to learn our timbre transfer models, we rely on the Studio-On-Line (SOL) database of orchestral instrument note recordings (Ballet et al. (1999)). We selected 12 instruments across the 4 families of *wind* (Alto-Saxophone, Bassoon, Clarinet, Flute, Oboe), *brass* (English-Horn, French-Horn, Tenor-Trombone, Trumpet), *string* (Cello, Violin) and *keyboard* (Piano). We consider each instrumental subset as a timbral domain $\mathcal{X}_i$, which contains the full tessitura of each instrument at different velocities (amounting to around 100 to 200 samples per domain). We split these subsets into 90% training notes and 10% test set. The audio waveforms are down-sampled to 22050Hz before computing the Non-Stationary Gabor Transform (NSGT) (Balazs et al. (2011)). This spectral transform allows to map to a perceptual pitch scale, while remaining iteratively invertible to the signal domain (Perraudin et al. (2013)). NSGTs are computed on a scale of 500 Mel bins ranging from 10Hz to 11000Hz. The resulting matrix data is sliced into chunks of 16 temporal frames, amounting for a context of about 120ms. This yields a final input size of 16x500 dimensions. We keep only the magnitude information and lowest values are floored to $6e^{-5}$ before applying a logarithmic transform. Finally, we normalize the entire dataset by computing a zero-mean unit-range normalization on all training samples.

**Implementation details.** The *one-to-one* transfer network is implemented as follows. The first encoding stacks $e_d$ are domain-specific, each composed of two 2-dimensional strided convolutions, an intermediate flattening step and a fully-connected (FC) layer. The intermediate representation is either concatenated (denoted as CAT-po models) with the conditioning vector of size 21 (12 pitch and 9 octave classes) or modulated by FiLM (denoted as FiLM-po models). Follows a weight-shared set $e_{ws}$ of two FC layers and two Gaussian encoder output layers mapping the input to a shared 3-dimensional latent space. All intermediate layers are followed by batch normalization and a Leaky-ReLU non-linearity. This structure is mirrored in the decoders, where the latent code is conditioned and fed into a weight-shared block $d_{ws}$ of 3 FC layers. Follow separate decoding stacks $d_d$ each with a FC layer, two transpose convolutions that up-sample the representation and two Gaussian decoder outputs. The final output activation is a Tanh applied to the decoded means, according to the initial data scaling. Full details of the architectures are given in appendix A.

The *many-to-many* transfer model relies on the same architecture, but without domain-specific encoders and decoders. Hence, all layers are similar but a single network jointly processes all domains thanks to FiLM layers (denoted as FiLM-poi models). For these, an embedding layer maps our categorical vocabulary of pitch, octave and instrument classes to dense vectors which are processed by two FiLM generators, one for the encoder and one for the decoder. Each has 3 FC layers followed by scaling and biasing output pairs that each map to the size of the modulated layer. We replace all batch normalization by instance normalization and apply FiLM generator outputs as linear transform modulating normalized hidden activations. Conditional instance normalization is performed feature-wise for 1-dimensional vectors and channel-wise for 2-dimensional feature maps.

Regarding optimization, all training objectives are simultaneously back-propagated throughout all networks. We use a Xavier weight initialization and the ADAM optimizer with an initial learning rate of $1e^{-4}$. Following the $\beta$-warmup procedure (Sønderby et al. (2016)), only the NLL reconstruction objective is optimized in the first epochs and the KLD strength is gradually increased from 0 to 1 during half the total number of training epochs. Similarly, we introduce the translation objective after 40 epochs and the optional circle-consistency objective after 60 epochs. We train on mini-batches of size 128 and the MMD is evaluated against batches of size 2048 sampled from the target distributions and computed with three Gaussian kernel parameter values $\{0.05, 0.1, 1\}$. We found the magnitude of MMD gradients to be much smaller than that of the ELBO. Hence, we set $\lambda_{MMD}$ to $1e^5$. Given that our models are light, the training over instrument pairs or triplets can be done in less than 24 hours on a single mid-range GPU (eg. NVIDIA TITAN Xp 12Gb).

## 4.1 ONE-TO-ONE TRANSFER

First, we compare our MoVE proposal to UNIT on the one-to-one transfer task. To do so, we learn a different model for each pair of instruments. We perform incremental comparisons by ablating certain aspects of our proposal to assess their importance. First, we add concatenative conditioning of pitch and octave to UNIT (noted UNIT(GAN;C-po)). Then we add our proposed alternative MMD criterion replacing the GAN objective (UNIT(MMD;C-po)). Then, we introduce the FiLM layers leading to our MoVE proposal. The first version still features separate domain-specific encoders and decoders, so it is noted MoVE* (MMD; F-po). By further introducing domain conditioning and relying on a single VAE (as in Figure 2), we obtain our proposal MoVE (MMD; F-pod).

**Evaluation scores.** To evaluate reconstruction performances, we compute several criteria between input samples x and reconstructions $\tilde{x}$. The Root-Mean-Square Error $RMSE = \sqrt{\sum(x - \bar{x})^2}$ and Log-Spectral Distortion $LSD = \sqrt{\sum(10 * \log_{10}(\frac{x^2}{\bar{x}^2}))^2}$ provide different assessments of how various models are able to reconstruct samples from the test set. Therefore, they only assess reconstruction abilities without domain transfer. To evaluate the quality of domain transfers, we compute the Maximum Mean Discrepancy (MMD) and the non-differentiable k-Nearest Neighbour (k-NN) test (Friedman & Rafsky (1983)). Both are dissimilarity measures computed between the target data distribution and transferred samples. Hence, we evaluate test set transfers between different target domains.

**Reconstruction and domain transfer.** The averaged reconstruction and transfer results are presented in Table 3, while separate evaluations for different pairs are in Annex B. As we can see, the UNIT-MMD model obtains the highest within-domain reconstruction score, while the MoVE model achieves better domain translation. Hence, it appears that the MMD increases reconstruction

performance, and that the FiLM conditioning ameliorates the transfer. It also seems that relying on a single encoder and decoder for domain transfer might provide better generalization, as can be verified by looking at the relative MMD and kNN scores on the transfer task. Indeed, it seems that the modulated but separate layers approach perform worse, while the single architecture performs better on most evaluations.

Table 1: Evaluations of various models on the test sets

| | reconstructions | | | | transfers | |
|---|---|---|---|---|---|---|
| | RMSE | LSD | MMD ($\alpha = 0.05$) | k-NN (k = 10) | MMD ($\alpha = 0.05$) | k-NN (k = 10) |
| UNIT (GAN) | 0.3412 | 718.47 | 2.117e-2 | 57269 | 2.038 e-2 | 43180 |
| UNIT (GAN; C-po) | **0.3011** | 693.22 | 1.989 e-2 | 57806 | 9.112 e-2 | 43414 |
| UNIT (MMD; C-po) | 0.3036 | **692.41** | 2.125 e-2 | **57102** | 2.304 e-2 | 43878 |
| MoVE* (MMD; F-po) | 0.3134 | 762.51 | 9.632 e-3 | 57273 | 3.153 e-2 | 43443 |
| MoVE (MMD; F-pod) | 0.3339 | 781.11 | **2.587 e-3** | 57509 | **1.747 e-2** | **43173** |

**Audio descriptors topology.** Audio descriptors are features used to compare the qualities of different sounds (Peeters et al. (2011)). Hence, we rely on these to assess the effect of transfer, while providing a deeper understanding of its behavior. We compute the spectral *flatness*, *centroid*, *roll-off* and *loudness* on test samples reconstructed on their own domain or transferred to the other domain. Distribution and sample-specific plots for the spectral centroid are presented in Figure 3.

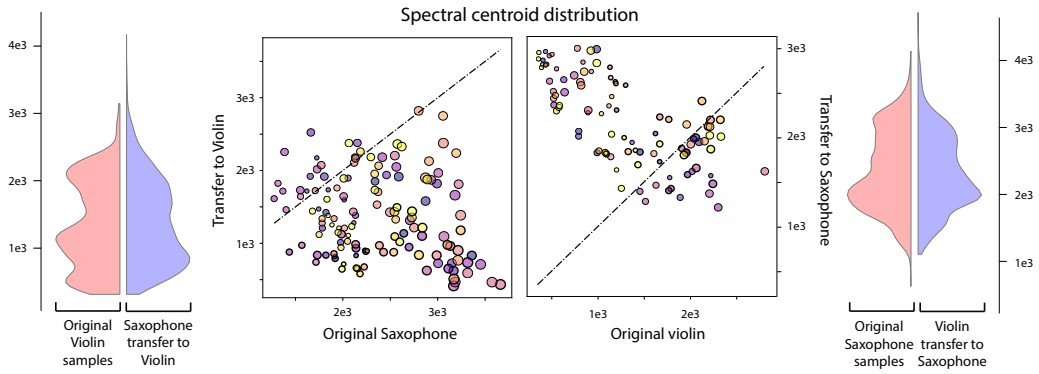

Figure 3: Understanding the effect of musical timbre transfer through audio descriptor distributions.

As we can see, the transfer produces an almost exact match of the descriptor distribution to the target domain. This shows the success in transferring multimodal distributions of auditory properties, as all the modes of the descriptors' distributions are preserved. The scatter plot also suggests that the centroid transfer is highly influenced by the loudness of the sample. This correlates to perception studies, as playing an instrument louder usually leads to a higher centroid McAdams et al. (1995).

In order to further understand how the latent space is organized with respect to audio descriptors, we provide their spatial topology in Figure 4. To compute this, we define a sampling grid over the latent space and decode the audio at each point to compute their descriptors. As we can see, the audio descriptors are locally very smooth. Furthermore, one key observation is that the latent space of both conditioned target domains follow the same overall topology. Animations showing the complete latent descriptor topology are available on the supporting webpage.

**Latent space synthesis and performance.** As the latent space provides continuous audio synthesis and that our method introduce high-level conditioned controls, we can use our proposal as a full musical synthesizer. Furthermore, as we map to 3-dimensional spaces, the user can directly interact with the space while performing timbre transfer. Furthermore, although these models are trained to

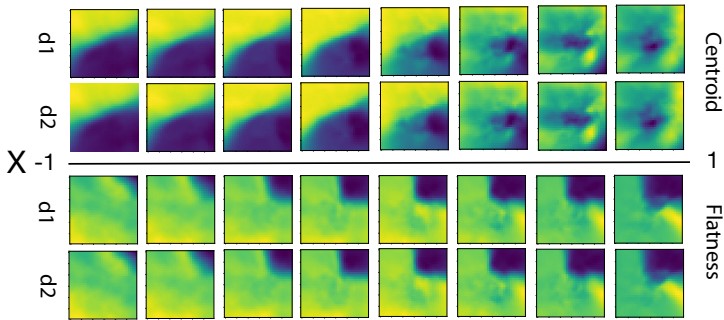

Figure 4: Topology of the latent space with respect to audio descriptors.

transfer single instrumental notes, they still can be used to transfer a full melody recording between timbre domains. To do so, the recording is split and iteratively reconstructed by transfering each signal window to the target domain. Audio examples of applying this strategy to transfer a complete instrumental solo are also available in the supporting webpage.

### 4.2 MANY-TO-MANY TRANSFER

Here, we evaluate the application of MoVE to perform many-to-many transfer. Given our new architecture, this simply consists in training on multiple domains at once by modulating with the appropriate domain information. This architecture allows us to train a single model for different domains and thus to perform multi-domain translation. The conditioning vector is then composed of the pitch, the octave and here the instrument of the corresponding example. This conditioning vector is then processed by an embedding to ease the FILM conditioning. Results are presented in Table 2 : we can see that the MoVE architecture is able to reconstruct and transfer multiple domains at the same time at the cost of a slight decrease in performance, even in the case of diverse domains (here Alto-Sax, Flute, Violin and French-Horn).

Table 2: Many-to-Many MoVE reconstruction & transfer scores

|  | averaged reconstructions | | | | averaged transfers | |
|---|---|---|---|---|---|---|
|  | RMSE | LSD | MMD ($\alpha = 0.05$) | k-NN (k = 10) | MMD ($\alpha = 0.05$) | k-NN (k = 10) |
| Alto-Saxophone | 0.5327 | 835.67 | 2.117 e-2 | 42299 | 2.386 e-2 | 59157 |
| Flute | 0.4593 | 761.46 | 2.119 e-2 | 49719 | 1.975 e-2 | 57277 |
| Violin | 0.3271 | 773.65 | 5.659 e-3 | 58013 | 1.452 e-2 | 55379 |
| French-Horn | 0.6239 | 869.69 | 3.404 e-3 | 70946 | 2.086 e-2 | 51317 |

## 5 CONCLUSION

We introduced the Modulated Variational auto-Encoders (MoVE), which perform many-to-many domain transfer within a single architecture and without adversarial training while providing high-level control over the generation. We effectively adapted this technique to musical timbre transfer and showed the successes of our method for audio synthesis. As our technique is generic, it could be applied to other types of data such as image or video. The architecture itself opens up a range of potential sonic applications such as playing style conditioning, transfers between acoustical and electronic instruments, and even with non-musical sound domains. Another avenue of research to be investigated is controlling the amount of transfer performed by the model.

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

## APPENDIX A - ARCHITECTURE DETAILS

Detailed layer definitions are given according to the following nomenclature, by default all learnable output biases are trained.

**2-dimensional strided convolutions =**
conv [out channels, (2D kernel), (2D stride), (2D padding)]
**2-dimensional strided transpose convolutions =**
convT [out channels, (2D kernel), (2D stride), (2D padding), (2D output padding)]
**fully-connected layer =** FC [out features]
**batch normalization =** BN-1D or BN-2D for 1 or 2-dimensional
**instance normalization =** IN-1D or IN-2D for 1 or 2-dimensional
**hidden activation =** act for Leaky-ReLU
**embedding =** embed [output size]
**FiLM conditioning =** FiLM-1D or FiLM-2D for feature-wise or channel wise
linear transform layers from the FiLM generator outputs, size according to the [hidden shape]
**concatenative conditioning =** CAT [condition size]

**(2d)** indicates two domain-specific instances of the same layer
**(2g)** indicates two Gaussian output instances of the same layer

| encoders | UNIT (GAN; C-po) | MoVE* (MMD; F-po) | MoVE (MMD; F-pod) |
|---|---|---|---|
| Enc.0 | **(2d)** conv [32,(9,21),(3,3),(4,10)] | | conv [32,(9,21),(3,3),(4,10)] |
| Norm.E0 | BN-2D + act | IN-2D + act | IN-2D + act + FiLM-2D[E0] |
| Enc.1 | **(2d)** conv [64,(6,15),(1,3),(0,7)] | | conv [64,(6,15),(1,3),(0,7)] |
| Norm.E1 | BN-2D + act | IN-2D + act | IN-2D + act + FiLM-2D[E1] |
| | intermediate flattening | | |
| Enc.2 | **(2d)** FC [4096] | | FC [4096] |
| Norm.E2 | BN-1D + act + CAT [21] | IN-1D + act | |
| Enc.3 | FC [2048] | | |
| Norm.E3 | BN-1D + act | IN-1D + act + FiLM-1D[E3] | |
| Enc.4 | FC [1024] | | |
| Norm.E4 | BN-1D + act | IN-1D + act + FiLM-1D[E4] | |
| $\mu_z; \sigma_z$ | **(2g)** FC [3] | | |
| | sampling z $\sim \mathcal{N}(\mu_z, \sigma_z)$ | | |
| **decoders** | | | |
| Dec.0 | CAT [21] + FC [1024] | FC [1024] | |
| Norm.D0 | BN-1D + act | IN-1D + act + FiLM-1D[D0] | |
| Dec.1 | FC [2048] | | |
| Norm.D1 | BN-1D + act | IN-1D + act + FiLM-1D[D1] | |
| Dec.2 | FC [4096] | | |
| Norm.D2 | BN-1D + act | IN-1D + act | |
| Dec.3 | **(2d)** FC [64*56] | | FC [64*56] |
| Norm.D3 | BN-1D + act | IN-1D + act | |
| | intermediate unflattening to 64 channels | | |
| Dec.4 | **(2d)** convT [64,(6,15),(1,3),(0,7),(0,1)] | | convT [64,(6,15),(1,3),(0,7),(0,1)] |
| Norm.D4 | BN-2D + act | IN-2D + act | IN-2D + act + FiLM-2D[D4] |
| Dec.5 | **(2d)** convT [32,(9,15),(3,3),(4,7),(0,1)] | | convT [32,(9,15),(3,3),(4,7),(0,1)] |
| Norm.D5 | BN-2D + act | IN-2D + act | IN-2D + act + FiLM-2D[D5] |
| $\mu_x; \sigma_x$ | **(2d*2g)** convT [1,(5,15),(1,1),(2,7),(0,0)] | | **(2g)** convT [1,(5,15),(1,1),(2,7),(0,0)] |
| | training: x $\sim \mathcal{N}(\text{TanH}(\mu_x), \sigma_x)$ generation: x $\sim \text{TanH}(\mu_x)$ | | |
| FiLM generators | | embed [21] FC [128] + IN-1D + act FC [512] + IN-1D + act FC [2048] + IN-1D + act | |
| FiLM encoder outputs FiLM decoder outputs | | FC [FiLM-1D[E3]+FiLM-1D[E4]] FC [FiLM-1D[D0]+FiLM-1D[D1]] | FC [FiLM-1D[E0]+FiLM-1D[E1]+FiLM-1D[E3]+FiLM-1D[E4]] FC [FiLM-1D[D0]+FiLM-1D[D1]+FiLM-1D[D4]+FiLM-1D[D5]] |

## Appendix B - Detailed transfer evaluations

Following scores of the table 3 are averaged on the pairs Alto-Saxophone+Violin ; Flute+French-Horn ; Oboe+Cello and Clarinet+Piano.

Table 3: Averaged scores for reconstruction and transfer over the test set samples

|  | averaged reconstructions | | | | averaged transfers | |
|---|---|---|---|---|---|---|
|  | RMSE | LSD | MMD ($\alpha = 0.05$) | k-NN (k = 10) | MMD ($\alpha = 0.05$) | k-NN (k = 10) |
| UNIT (GAN; C-po) | **0.4083** | **701.06** | 1.7223 e-2 | 59499 | 1.955 e-2 | 60269 |
| MoVE* (MMD; F-po) | 0.4405 | 788.09 | 1.244 e-2 | 59300 | 1.650 e-2 | **59975** |
| MoVE (MMD; F-pod) | 0.4300 | 779.25 | **1.007 e-2** | **59282** | **1.487 e-2** | 60455 |

Table 4: Pair-wise evaluations on Flute and French-Horn test sets

|  | reconstructions Flute | | | | transfers to French-Horn | |
|---|---|---|---|---|---|---|
|  | RMSE. | LSD. | MMD. ($\alpha = 0.05$) | k-NN. (k = 10) | MMD. ($\alpha = 0.05$) | k-NN. (k = 10) |
| UNIT (GAN; C-po) | 0.2773 | **680.49** | **3.425 e-3** | 50344 | **2.991 e-4** | 71596 |
| MoVE* (MMD; F-po) | 0.3724 | 792.67 | 1.967 e-2 | 49840 | 6.375 e-3 | 71582 |
| MoVE (MMD; F-pod) | **0.2697** | 698.68 | 2.226 e-2 | 49945 | 2.364 e-2 | 71848 |
|  | reconstructions French-Horn | | | | transfers to Flute | |
| UNIT (GAN; C-po) | **0.5442** | **736.96** | **1.942 e-4** | 71672 | **8.888 e-3** | 52304 |
| MoVE* (MMD; F-po) | 0.5749 | 800.64 | 3.458 e-3 | 72029 | 4.156 e-2 | **51250** |
| MoVE (MMD; F-pod) | 0.6026 | 820.71 | 2.584 e-3 | 71650 | 2.049 e-2 | 52348 |

Table 5: Pair-wise evaluations on Oboe and Cello test sets

|  | reconstructions Oboe | | | | transfers to Cello | |
|---|---|---|---|---|---|---|
|  | RMSE. | LSD. | MMD. ($\alpha = 0.05$) | k-NN. (k = 10) | MMD. ($\alpha = 0.05$) | k-NN. (k = 10) |
| UNIT (GAN; C-po) | **0.4928** | **686.52** | 8.196 e-3 | 45660 | 2.238 e-2 | **61267** |
| MoVE* (MMD; F-po) | 0.5641 | 808.06 | **2.093 e-3** | 45705 | **3.290 e-3** | 62272 |
| MoVE (MMD; F-pod) | 0.5509 | 776.80 | 3.440 e-3 | 45378 | 7.486 e-3 | 62036 |
|  | reconstructions Cello | | | | transfers to Oboe | |
| UNIT (GAN; C-po) | **0.2784** | **711.37** | 9.579 e-3 | 61775 | 6.394 e-3 | 47548 |
| MoVE* (MMD; F-po) | 0.3226 | 777.27 | 2.093 e-3 | 61736 | **2.428 e-3** | **47250** |
| MoVE (MMD; F-pod) | 0.3487 | 818.90 | **1.769 e-3** | 61952 | 8.352 e-3 | 47787 |

Table 6: Pair-wise evaluations on Clarinet and Piano test sets

| | reconstructions Clarinet | | | | transfers to Piano | |
|---|---|---|---|---|---|---|
| | RMSE. | LSD. | MMD. ($\alpha = 0.05$) | k-NN. (k = 10) | MMD. ($\alpha = 0.05$) | k-NN. (k = 10) |
| UNIT (GAN; C-po) | 0.6915 | **778.33** | 3.918 e-2 | 76035 | 2.827 e-2 | 74273 |
| MoVE* (MMD; F-po) | 0.7351 | 916.88 | **1.144 e-2** | **75440** | **1.287 e-2** | **73753** |
| MoVE (MMD; F-pod) | **0.6878** | 884.63 | 1.614 e-2 | 75604 | 1.462e-2 | 74349 |
| | reconstructions Piano | | | | transfers to Clarinet | |
| UNIT (GAN; C-po) | 6.682 e-2 | **543.09** | 2.610 e-2 | 70925 | 4.071 e-2 | 74268 |
| MoVE* (MMD; F-po) | 6.522 e-2 | 582.98 | **1.276 e-2** | 70055 | **6.790 e-3** | **72761** |
| MoVE (MMD; F-pod) | **5.727 e-2** | 578.66 | 1.547 e-2 | **70044** | 1.884 e-2 | 74801 |

Table 7: *many-to-many* MoVE evaluation on Clarinet
Cello, Tenor-Trombone (T-Trombone) and Piano test sets (350 epochs only)

| input | reconstructions | | | | targets | transfers | |
|---|---|---|---|---|---|---|---|
| | RMSE. | LSD. | MMD. ($\alpha = 0.05$) | k-NN. (k = 10) | | MMD. ($\alpha = 0.05$) | k-NN. (k = 10) |
| Clarinet | 0.6832 | 873.17 | 9.776 e-3 | 76093 | Cello | 1.205 e-2 | 64117 |
| | | | | | T-Trombone | 7.450 e-3 | 29896 |
| | | | | | Piano | 0.1233 | 75278 |
| Cello | 0.3043 | 755.58 | 2.115 e-3 | 61756 | Clarinet | 1.345 e-3 | 74569 |
| | | | | | T-Trombone | 1.326 e-3 | 28632 |
| | | | | | Piano | 0.1595 | 74420 |
| T-Trombone | 0.7362 | 1047.1 | 1.281 e-2 | 25629 | Clarinet | 2.089 e-2 | 73320 |
| | | | | | Cello | 1.652 e-2 | 61006 |
| | | | | | Piano | 0.1649 | 71939 |
| Piano | 4.965 e-2 | 542.72 | 9.773 e-3 | 69573 | Clarinet | 0.1482 | 75589 |
| | | | | | Cello | 0.1155 | 63947 |
| | | | | | T-Trombone | 0.2038 | 28675 |

