# OpenReview forum: "Modulated Variational Auto-Encoders for Many-to-Many Musical Timbre Transfer"
_ICLR.cc/2019/Conference_

### Official Review · AnonReviewer2 · 2018-11-01
**Nice idea but falls short of what it promises**

**Rating:** 3
**Confidence:** 4

**Review:**

This work proposes a hybrid VAE-based model (combined with an adversarial or maximum mean discrepancy (MMD) based loss) to perform timbre transfer on recordings of musical instruments. Contrary to previous work, a single (conditioned) decoder is used for all instrument domains, which means a single model can be used to convert any source domain to any target domain.

Unfortunately, the results are quite disappointing in terms of sound quality, and feature many artifacts. The instruments are often unrecognisable, although with knowledge of the target domain, some of its characteristics can be identified. The many-to-many results are clearly better than the pairwise results in this regard, but in the context of musical timbre transfer, I don't feel that this model successfully achieves its goal -- the results of Mor et al. (2018), although not perfect either, were better in this regard.

I have several further concerns about this work:

* The fact that the model makes use of pitch class and octave labels also raises questions about applicability -- if I understood correctly, transfer can only be done when this information is present. I think the main point of transfer over a regular generative model that goes from labels to audio is precisely that it can be done without label information.

* The use of fully connected layers also implies that it requires fixed length input, so windowing and stitching are necessary for it to be applied to recordings of arbitrary length. Why not train a convolutional model instead?

* I think the choice of a 3-dimensional latent space is poorly justified. Why not use more dimensions and project them down to 3 for visualisation and interpetation purposes with e.g. PCA or t-SNE? This seems like an unnecessary bottleneck in the model, and could partly explain the relatively poor quality of the results.

I appreciated that the one-to-one transfer experiments are incremental comparisons, which provides valuable information about how much each idea contributes to the final performance.

Overall, I feel that this paper falls short of what it promises, so I cannot recommend acceptance at this time.



Other comments:

* In the introduction, an adversarial criterion is referred to as a "discriminative objective", but "adversarial" (i.e. featuring a discriminator) and "discriminative" mean different things. I don't think it is correct to refer to an adversarial criterion as discriminative.

* Also in the introduction, it is implied that style transfer constitutes an advance in generative models, but style transfer does not make use of / does not equate to any generative model.

* Some turns of phrase like "recently gained a flourishing interest", "there is still a wide gap in quality of results", "which implies a variety of underlying factors", ... are vague / do not make much sense and should probably be reformulated to enhance readability.

* Introduction, top of page 2: should read "does not learn" instead of "do not learns".

* Mor et al. (2018) do actually make use of an adversarial training criterion (referred to as a "domain confusion loss"), contrary to what is claimed in the introduction.

* The claim that training a separate decoder for each domain necessarily leads to prohibitive training times is dubious -- a single conditional decoder would arguably need more capacity than each individual separate decoder model. I think all claims about running time should be corroborated by controlled experiments.

* I think Figure 1 is great and helps a lot to distinguish the different domain translation paradigms.

* I found the description in Section 3.1 a bit confusing as it initially seems that the approach requires paired data (e.g. "matching samples").

* Section 3.1, "amounts to optimizing" instead of "amounts to optimize"

* Higgins et al. (2016) specifically discuss the case where beta in formula (1) is larger than one. As far as I can tell, beta is annealed from 0 to 1 here, which is an idea that goes back to "Generating Sentences from a Continuous Space" by Bowman et al. (2016). This should probably be cited instead.

* "circle-consistency" should read "cycle-consistency" everywhere.

* MMD losses in the context of GANs have also been studied in the following papers:
- "Training generative neural networks via Maximum Mean Discrepancy optimization", Dziugaite et al. (2015)
- "Generative Models and Model Criticism via Optimized Maximum Mean Discrepancy", Sutherland et al. (2016)
- "MMD GAN: Towards Deeper Understanding of Moment Matching Network", Li et al. (2017)

* The model name "FILM-poi" is only used in the "implementation details" section, it doesn't seem to be referred to anywhere else. Is this a typo?

* The differences between UNIT (GAN; C-po) and UNIT (MMD; C-po) in Table 1 seem very small and I'm not convinced that they are significant. Why does the MMD version constitute an improvement? Or is it simply more stable to train?

* The descriptor distributions in Figure 3 don't look like an "almost exact match" to me (as claimed in the text). There are some clearly visible differences. I think the wording is a bit too strong here.

---

> ### Author Response · Authors · 2018-11-17
> **answer to the review**
>
> Thank you for the detailed review and constructive remarks.
> Below are answers to the main points that were commented as well as updates on the current work.
>
> * Sound quality is disappointing and with artifacts:
> We are working on Fast Spectrogram Inversion using Multi-head Convolutional Neural Networks, arXiv:1808.06719, Sercan O. Arik et al. to replace Griffin-Lim inversion ; two possible improvements we expect are much faster (towards real-time) sound rendering and better audio quality.
> We are also working on mini-batch MMD latent regularization (Wasserstein-AE) instead of per-sample KLD regularization (VAE) which may result in improved generalization power and generative quality.
>
> * Not suited to transfer from audio without label:
> If the audio carries a note information, it can be easily/automatically extracted in the form of pitch tracks as we did for transferring on instrument solos. Some audio data do not have note qualities, which are out of the current training setting. For that we have been training unconditioned one-to-one models or solely instrument conditional many-to-many models that do not require any note information.But we are working on models which incorporate an unconditioned processing option (eg. training while zeroing the one-hot conditioning or adding an entry in the input embedding of FiLM which is the unconditional state) to be trained on a dataset that mixes conditional and non conditional audio (eg. adding instrument solo sections which in parts have a clear pitch track and in others none).
>
> * A fully convolutional model would process arbitrary length of audio:
> We use the linear layers to set the latent space dimensionality, when processing various length audio sequences, each encoding amounts to about 120ms context and we resynthesize with overlap-ad that mirrors the short-term input analysis ; this process was used when transferring on the instrument solos (a task that was beyond the training setting).
>
> * Insufficient justification of the 3D latent space:
> At first we validated that our models could perform well in term of training/test spectrogram reconstructions with only 3 latent dimensions, some reasons that we found interesting to enforce this are more related to a possible music/creative application of the model: less synthesis/control parameters for the user (and controls which may then be more expressive), direct visualization of the latent space which is turned into a 3D synthesis space from which users may draw and decode sound paths or create other interaction schemes, a denser latent space that may be better suited for random sampling/interpolations. The direct interaction with 3D latent space becomes even more interesting when we pipeline our model with fast-spectrogram inversion.
>
> * Interesting incremental comparison in one-to-one transfers:
> We keep working on more detailed benchmarks/comparisons that would equally cover one-to-one and many-to-many model variations and that would integrate the new features we are testing.
>
> * All claims about running time should be corroborated by controlled experiments:
> Indeed we didn’t benchmark yet our models on Nsyth and our approach differs from others such as Mor et al. that report using « eight Tesla V100 GPUs for a total of 6 days ». From the beginning of our experiment we aim at a much lighter-weight system that could be trained/used more broadly (eg. with a single mid-range GPU). The computational cost difference is not rigorously estimated on a same given dataset/task to learn but still we think it is relevent to point that the results we report can be achieved in less that a day on a single Tesla V100 GPU.
>
> * Why does the MMD version constitute an improvement? Or is it simply more stable to train?
> It is more stable to train, it does not require the extra ‘cost’ of an auxiliary network training and it can generalize to many-to-many transfer without requiring as many adversarial networks. About the significance of score differences, we agree that it needs more details and comparisons, it was also noted by "AnonReviewer1" and we should make alternative tests to scale or give a few more references to the benchmark.
>
> * "FILM-poi" .. is this a typo ?
> Thank you for pointing this as well as your other remarks on the writing and use of precise terms/phrases. Indeed this is right, we mixed poi/pod but both refer to many-to-many conditioning on pitch+octave+instrument/domain classes.
>
> We also thank you for pointing more literature to improve our references and discussions to related works.

---

> > ### Comment · AnonReviewer2 · 2018-11-21
> > **response to authors' comments**
> >
> > Thank you. I have a few more comments:
> >
> > Regarding pitch label extraction: if this is necessary anyway, then why not just train a pitch-conditional generative model? What is the benefit of additionally conditioning the model on the original audio, in this case? I still think it defeats the purpose of the "transfer" aspect a bit. Have you assessed at all how much the model actually relies on being conditioned on the original audio? Perhaps it is already largely ignoring it, and just using the pitch label to know what to generate.
> >
> > Regarding the 3D latent space: this is a reasonable argumentation and it would be good to make this more clear in the paper itself. Still, some comparison experiments with different latent space dimensionalities would be useful to demonstrate that 3 is enough.
> >
> > Regarding claims about running time: it is definitely worth stating that the results you report can be achieved in less than a day on a single Tesla V100 GPU. But the claim in the paper was specifically about other strategies than the chosen one taking much longer, and this is not corroborated, so it would be good to reformulate this.

---

> > > ### Author Response · Authors · 2018-11-24
> > > **answer to the reviewer comment**
> > >
> > > Thank you for the further discussion.
> > >
> > > So far our models were either non-conditional or conditional on both encoder and decoder.
> > > Models without the pitch information could train well but did not generalized so well.
> > > They tended to reconstruct with some transposition or on wrong octave.
> > >
> > > I am considering on experimenting with having a non-conditional state inside the conditional models.
> > > I am also considering conditioning only the decoder. If that is what you refer as only having pitch-conditional generative model. It would indeed offer a larger application potential at encoding unlabelled audio while offering the generative control.
> > >
> > > Regarding the latent space dimensionality, I will add more details on the impact of lowering down to 3. I as well worked on an alternative WAE-MMD regularization instead of KLD. Which would be more flexible, the lowered performance of the 3D latent space may be balanced with a larger prior variance or a more efficient prior choice.
> > >
> > > I will improve my experiment, based on the constructive remarks I receive from you and from AnonReviewer1. And will submit the newer version again to later conferences.

---

### Official Review · AnonReviewer1 · 2018-11-02
**Interesting and well written, but the evaluation is difficult to interpret**

**Rating:** 5
**Confidence:** 3

**Review:**

Summary
-------
This paper describes a model for musical timbre transfer which builds on recent developments in domain- and style transfer.
The proposed method is designed to be many-to-many, and uses a single pair of encoders and decoders with additional conditioning inputs to select the source and target domains (timbres).
The method is evaluated on a collection of individual note-level recordings from 12 instruments, grouped into four families which are used as domains.
The method is compared against the UNIT model under a variety of training conditions, and evaluated for within-domain reconstruction and transfer accuracy as measured by maximum mean discrepancy.
The proposed model seems to improve on the transfer accuracy, with a slight hit to reconstruction accuracy.
Qualitative investigation demonstrates that the learned representation can approximate several coarse spectral descriptors of the target domains.


High-level comments
-------------------
Overall, this paper is well written, and the various design choices seem well-motivated.

The empirical comparisons to UNIT are reasonably thorough, though I would have preferred more in-depth evaluation of the MoVE model as well.  Specifically, the authors introduced an extra input (control) to encode the pitch class and octave information during encoding.  I infer that this was necessary to achieve good performance, but it would be instructive to see the results without this additional input, since it does in a sense constitute a form of supervision, and therefore limits the types of training data which can be used.

While I understand that quantifying performance in this application is difficult, I do find the results difficult to interpret.  Some of this comes down to incomplete definition of the metrics (see detailed comments below).
However, the more pressing issue is that evaluation is done either sample-wise within-domain (reconstruction), or distribution-wise across domains (transfer). The transfer metrics (MMD and kNN) are opaque to the reader: for instance, in table 1, is a knn score of 43173 qualitatively different than 43180?  What is the criteria for bolding here?  It would be helpful if these scores could be calibrated in some way, e.g., with reference to
MMD/KNN scores of random partitions of the target domain samples.

Since the authors do additional information here for each sample (notes), it would be possible to pair generated and real examples by instrument and note, rather than (in addition to) unsupervised, feature-space pairing by MMD.  This could provide a slightly stronger version of the comparison in Figure 3, which shows that the overall distribution of spectral centroids is approximated by transfer, but does not demonstrate per-sample correspondence.



Detailed comments
-----------------
At several points in the manuscript, the authors refer to "invertible" representations (e.g., page 4, just after eq. 1), but it seems like what they mean is approximately invertible or decodable.  It would be better if the authors were a little more careful in their use of terminology here.

In the definition of the RBF kernel (page 4), why is there a summation?
 What does this index? How are the kernel bandwidths defined?

How exactly are reconstruction errors calculated: using the NSGT magnitude representation, or after resynthesis in the time domain?

---

> ### Author Response · Authors · 2018-11-17
> **answer to the review**
>
> Thank you for your detailed review and the constructive comments on our work. We note the remarks on the paper writing that we will correct and answer below the main points that were commented.
>
> * In-depth evaluation of MoVE and comparison of with/without conditioning:
> We agree and this was also pointed by 'AnonReviewer2', we are working on new incremental benchmarks, more detailed on both one-to-one and many-to-many models. Moreover, the need of pitch/octave conditioning limits the applicability of our model to transfer only on audio carrying such note features. Hence we trained models without conditioning mechanism and, as answered to 'AnonReviewer2', we are planning experiments on models which are conditional but integrating an unconditioned state to be trained in parallel of the note-conditional state.
>
> *** Interpretability of the generative scores:
> We agree on this remark, the idea of scaling scores is right and would improve the interpretability of our benchmarks. For that purpose, we should define a set of reference scores as you recommended to.
>
> * Incomplete definition of the metrics:
> We gave references to the papers that introduced such metrics. Discussing a set of reference scores should also come with a better explanation of these.
>
> * Criteria for bolding: we intended to highlight the best scores
>
> *** Pairing generated and real examples by instrument and note to compare:
> In addition to the spectral descriptor distribution plots, we used sample-specific scatter plots to visualize how the transfer maps them individually. On the overlap of each instrument tessitura, we can make such pairing. We can also transfer and transpose to the target instrument tessitura if needed. Remains the question of which metric can be used here to evaluate generation at the sample-level (?), as our model does not aim at reconstructing an hypothetical corresponding sample in the target domain but rather at blending in features from the other domain so that it sounds like the input note (pitch, octave but also some dynamics/style qualities relative to the input instrument) played by the target instrument. We later aim at experimenting on mechanisms to control the amount of target feature blending in the process of transfer.
>
> * Invertible ? Decodable ? Approximate inversion ?
> We agree that the current state of the research should be stated as using approximate spectrogram inversion.
> We plan on replacing the iterative slow spectrogram inversion with Griffin-Lim by faster decoding with Multi-head Convolutional Neural Networks, arXiv:1808.06719, Sercan O. Arik et al.
>
> *** Definition of the RBF kernel:
> The summation is on the alpha parameter which can be a list of n values (or a single float value). The trainings were done with n=3 and alpha=[1. , 0.1 , 0.05]. Depending on the kernel and bandwidth definitions, we may link both as
> alpha = 1 / (2 x bandwidth**2).
>
> * Calculation of reconstruction errors:
> All scores are computed on NSGT magnitude spectrogram slices. No evaluation (except listening) is done on the time-domain waveforms.
>
> The points marked with *** are highlighted as we would gratefully receive further remarks from your review.
> How would you recommend making reference scores to the MMD/kNN evaluations ?
> How would you recommend comparing pairs of generated and ~ corresponding target domain samples ? (at the sample level)
> Is the definition of the RBF kernel correct to you given that clarification (that should be added to the paper) ?
>
> Thanks again for the interesting feedbacks !

---

> > ### Comment · AnonReviewer1 · 2018-11-29
> > **Re: author response**
> >
> > Re: Scores and bolding
> >
> > It's clear that you intend to bold the "best" score, but sorting by mean RMSE (or whatever metric you choose) isn't particularly robust, since it ignores the variance of each method.  I'm skeptical that differences in the third decimal place are really significant, and some kind of statistical test for significance in distinguishing from the "best" seems warranted here (paired t-test, wilcoxon sign-rank, something).
> >
> > For the MMD and KNN metrics, as I suggested initially, you might look at the scores produced by generating random partitions of samples from the target distribution.  This would at least give a lower bound on the scores you could hope to achieve with a good approximation of the target distribution.
> >
> >
> > Re: paired comparisons
> >
> > As I said initially, the current evaluation (eg Fig. 3) looks at the distribution of (features of) samples from the different instruments (transferred and real target).  But, since you also have pitch and octave information, all I meant was that you could do a more specific comparison that is restricted to the same pitch / octave.   This would demonstrate whether the system is approximating the distribution on average, or sample by sample.
> >
> >
> > Re: RBF kernel, your definition seems to be a sum of RBF kernels with different bandwidths, not an RBF kernel properly.  This should be clarified in the text.

---

> > > ### Author Response · Authors · 2018-11-30
> > > **answer to the review**
> > >
> > > thank you for the constructive remarks
> > >
> > > according to it, we are reconsidering how we compute scores and particularly scaling transfer objectives (MMD,KNN,Energy Statistics) with scores on different partitions of source/target domains against reference batches of the target domain
> > >
> > > we may also consider some sample-to-sample evaluations restricted to common tessitura
> > > eg. transferring a note from a domain to an other and comparing with the target domain sample that has the same note class
> > >
> > > it is correct also that you point that the formula is for a mix of RBF kernels with different bandwidth

---

### Official Review · AnonReviewer3 · 2018-11-07
**Interesting but is hardly to ready due to the confusing introduction**

**Rating:** 5
**Confidence:** 3

**Review:**

The authors proposed a Modulated Variational auto-Encoders (MoVE) to perform musical timbre transfer. The authors define timbre transfer as applying parts of the auditory properties of a musical instrument onto another. It replaces the usual adversarial translation criterion by a Maximum Mean Discrepancy (MMD) objective. By further conditioning our system on several different instruments, the proposed method can generalize to many-to-many transfer within a single variational architecture able to perform multi-domain transfers.
Some detailed comments are listed as follow,
1 The implementation steps of the proposed method (MoVE) are not clear. Some details are missing, which is hardly reproduced by the other researchers.
2 The experimental settings are not reasonable. The current experimental settings are not matched with the practice environment.
3 The proposed method can transfer the positive knowledge. However, some negative knowledge information can be also transferred. So how to avoid the negative transferring?
4 For the model, the optimization details or inferring details are missing, which are important for the proposed model.

---

> ### Author Response · Authors · 2018-11-21
> **answer to the review**
>
> Thank you for your review, below we answer the points that were questioned.
>
> * Missing implementation steps and optimization details:
> In addition to implementation details, the appendix has a rather detailed table of the architecture parameters. Moreover, we will ultimately release codes on Github.
>
> * Non-matched experiment to practice environment:
> The evaluation of generative models and unsupervised domain translations remains an open question, even less covered in the field of sound. We didn't apply our models yet to datasets previously covered in the related works, such as Nsynth, which is planned and would give some more direct comparisons.
>
> * How to avoid the negative knowledge transfer:
> As we defined our purpose, the resulting generation is a blending of both domains that renders a target timbre while retaining some of the input features. It amounts to note class (that is explicitly controlled for the note-conditional model states) together with timbre. We plan on experiments on controlling the amount of timbre transfer in between the input and target domains.

---

### Public Comment · (anonymous) · 2018-10-02
**The github repo is empty**

It seems that the github repo is empty.

---

> ### Author Response · Authors · 2018-10-03
> **Github repo under construction**
>
> (EDIT: some examples of transfer have been uploaded for the in-between time, you may have a look at the solo_transfers directory in the repository and hear a Violin solo transferred to Alto-Saxophone and reversely an Alto-Saxophone solo transferred to Violin)
>
> This is right, the Github repo is empty at the moment.
>
> We are currently working on the content and codes, by the end of next week we will have prepared and put online most of the audio examples, demonstrations and visualisations. Codes and new results will follow.
>
> Thank you for pointing it out, we invite you to visit the repo later again.

---

### Public Comment · (anonymous) · 2018-10-04
**extra bibliography**

Very detailed and impressive work !

I would just like to point out the previous series of work from Ilya Tolstikhin, Olivier Bousquet et al. on the use of MMD penalizations for both GAN or auto-encoders approaches, which could probably be added after the reference to Gretton et al.'s work, or rather as a comparative alternative to GANs (although your use differs greatly from theirs), see [1], [2] or [3] for instance.

[1] From optimal transport to generative modeling: the VEGAN cookbook (Bousquet et al., 2017)
[2] Wasserstein auto-encoders (Tolstikhin et al., 2018) - presented at ICLR 2018
[3] On the Latent Space of Wasserstein Auto-Encoders (Rubenstein et al., 2018)

---

> ### Author Response · Authors · 2018-10-08
> **thanks**
>
> Thank you for the positive and constructive feedback !
> Indeed, WAEs are of interest even though I only worked with the KLD latent regularization so far (here the MMD is applied to data distributions rather than latents as in WAEs).
> I didn't mention them but I agree that it would be better to at least add them to the related works and references.
>
> The main advantage reported compared to VAEs is their less blurry outputs in the case of image generation.
> It might also benefit to sound synthesis, but might be less crucial as spectrogram inversion is applied to the raw network outputs, which itself already introduces some posterior approximations (and audio artifacts).
>
> About this, for later experiments, I may consider replacing the use of Griffin-Lim by neural network spectrogram inversion, possibly using Wavenet Vocoder [1] or MCNN [2].
>
> [1] Wei Ping et al. "DEEP VOICE 3: SCALING TEXT-TO-SPEECH WITH CONVOLUTIONAL SEQUENCE LEARNING"
> [2] Sercan Arik et al. "Fast Spectrogram Inversion using Multi-head Convolutional Neural Networks"
>
> I wonder if this should also be added to the final paper (if selected).

---

### Author Response · Authors · 2018-10-18
**Repository progresses**

Hello everyone,
Since the submission, the repository has been developed.
Audio examples, visualisations and animations are detailed here:
https://github.com/anonymous124/iclr2019MoVE/blob/master/docs/index.md

Thank you for your interest, work is still ongoing and more will be uploaded throughout the next weeks.

---

### Meta-Review · Area_Chair1 · 2018-12-16
**Good motivation but important reviewers' concerns remain unaddressed**

**Confidence:** 5
**Recommendation:** Reject

**Metareview:**

This paper proposes a VAE-based model which is able to perform musical timbre transfer.

The reviewers generally find the approach well-motivated. The idea to perform many-to-many transfer within a single architecture is found to be promising. However, there have been some unaddressed concerns, as detailed below.

R3 has some methodological concerns  regarding negative transfer and asks for more extended experimental section.  R1 and R2 ask for more interpretable results and, ultimately, a more conclusive study. R2 specifically finds the results to be insufficient.

The authors have agreed with some of the reviewers' feedback but have left most of it unaddressed in a new revision. That could be because some of the recommendations require significant extra work.

Given the above, it seems that this paper needs more work before being accepted in ICLR.